# Experience-driven rate modulation is reinstated during hippocampal replay

Margot Tirole*†, Marta Huelin Gorriz*†, Masahiro Takigawa, Lilia Kukovska, Daniel Bendor*

Institute of Behavioural Neuroscience, University College London, London, United Kingdom

**Abstract** Replay, the sequential reactivation within a neuronal ensemble, is a central hippocampal mechanism postulated to drive memory processing. While both rate and place representations are used by hippocampal place cells to encode behavioral episodes, replay has been largely defined by only the latter – based on the fidelity of sequential activity across neighboring place fields. Here, we show that dorsal CA1 place cells in rats can modulate their firing rate between replay events of two different contexts. This experience-dependent phenomenon mirrors the same pattern of rate modulation observed during behavior and can be used independently from place information within replay sequences to discriminate between contexts. Our results reveal the existence of two complementary neural representations available for memory processes.

**\*For correspondence:**
margot.tirole.14@ucl.ac.uk (MT);
marta.huelin.16@ucl.ac.uk (MHG);
d.bendor@ucl.ac.uk (DB)

†These authors contributed equally to this work

**Competing interest:** The authors declare that no competing interests exist.

## Editor's evaluation

The hippocampal cells that comprise the place cell map for the most part 'remap' between different environments – they change their preferred firing locations and rates. This article poses an important question about offline reactivation that has not been explicitly tested: are differences in firing rate preserved during sequential, temporally compressed offline replay events? They find that yes, individual neurons show context-specific firing rates during replay, an important finding and a confirmation of critical theoretical foundations in the field of learning and memory. The evidence is convincing, with good support for the claims. At the same time, this demonstration hinges on some relatively subtle methodological points specific to replay detection, and thus serves as an invitation to the field to further explore the precise structure of context-specific offline activity.

## Introduction

A key function of the hippocampus is the initial encoding and subsequent consolidation of episodic memories. Hippocampal place cells are activated in sequential patterns during behavioral episodes (e.g., a rodent running toward a goal), with each neuron's activity modulated by the animal's position within an environment (i.e., a place field) (*OKeefe and Dostrovsky, 1971*). During offline periods such as sleep, place cells spontaneously reactivate, replaying the same sequential pattern previously activated during the behavioral episode (*Lee and Wilson, 2002*; *Wilson and McNaughton, 1994*). This phenomenon of neural replay is postulated to drive systems-level memory consolidation (*Crowley et al., 2019*; *Fernández-Ruiz et al., 2019*; *Girardeau et al., 2009*; *Gridchyn et al., 2020*). Hippocampal replay has largely been defined by the sequential reactivation of place cell ensembles (*Davidson et al., 2009*; *Foster and Wilson, 2006*; *Genzel et al., 2020*; *Lee and Wilson, 2002*), driven by both where and when each place field is activated during a behavioral episode (here referred to as place representation). Yet, place cells also carry additional information in the magnitude of their place field's activity (here referred to as rate representation). The magnitude of a place field's activity

**eLife digest** How do our brains store memories? We now know that this is a complex and dynamic process, involving multiple regions of the brain. A brain region, called the hippocampus, plays an important role in memory formation. While we sleep, the hippocampus works to consolidate information, and eventually creates stable, long-term memories that are then stored in other parts of the brain.

But how does the hippocampus do this? Neuroscientists believe that it can replay the patterns of brain activity that represent particular memories. By repeatedly doing this while we sleep, the hippocampus can then direct the transfer of this information to the rest of the brain for storage.

The behaviour of nerve cells in the brain underpins these patterns of brain activity. When a nerve cell is active, it fires tiny electrical impulses that can be detected experimentally. The brain thus represents information in two ways: which nerve cells are active and when (sequential patterns); and how active the nerve cells are (how fast they fire electrical impulses or firing rate). For example, when an animal moves from one location to another, special place cells in the hippocampus become active in a distinct sequence. Depending on the context, they will also fire faster or slower.

We know that the hippocampus can replay sequential patterns of nerve cell activity during memory consolidation, but whether it can also replay the firing rates associated with a particular experience is still unknown. Tirole, Huelin Gorriz et al. set out to determine if the hippocampus could also preserve the information encoded by firing rate during replay.

In the experiments, rats explored two different environments that they had not seen before. The activity of the rats' place cells was recorded before and after they explored, and also later while they were sleeping. Analysis of the recordings revealed that during replay, the rats' hippocampi could indeed reproduce both the sequential patterns of activity and the firing rate of the place cells. It also confirmed that each environment was associated with unique firing rates – in other words, the firing rates were memory-specific.

These results contribute to our understanding of how the hippocampus represents and processes information about our experiences. More broadly, they also shed new light on how the brain lays down memories, by revealing a key part of the mechanism that it uses to consolidate that information.

(i.e., peak in-field firing rate) is modulated by both local and global contextual cues (*Leutgeb et al., 2005*; *Ravassard et al., 2013*; *Zhao et al., 2020*), as well as behavioral variables (e.g., animal's speed during locomotion) (*Huxter et al., 2003*), and cognitive events (e.g., animal's attention and perception) (*Monaco et al., 2014*; *Olypher et al., 2002*). Reactivation events may also be capable of representing information using firing rate (*Schwindel et al., 2016*; *Takahashi, 2015*); however, because the detection of reactivation events is based on the firing rate of place cells, it is necessary to eliminate or at least control the effects of firing rate at the point of event detection to avoid any circular reasoning. Examining whether rate modulation is reinstated during replay will help discern whether replay only represents possible transitions between adjacent locations (i.e., the order of place fields within a spatial trajectory) or alternatively whether replay could serve a larger role, with the capacity to reinstate a near replica of neural activity previously driven during behavior, providing a richer, more complete memory trace.

## Results

### Place cells reinstate their rate modulation during replay

To address this question, we recorded extracellularly from the dorsal CA1 of rats (n = 5, male Lister-hooded), running back and forth along two novel linear tracks (2 m) to collect a liquid reward at each end (*Figure 1A*, *Figure 1—figure supplement 1A*, *Table 1*, and see 'Methods'). A view-obstructing divider was placed between tracks and distinguishing visual cues were present within each environment to facilitate contextual discrimination and, in turn, hippocampal remapping. Additionally, rats rested in a quiet remote location, both before (PRE) and after (POST) the exploration of the two tracks.

We observed global remapping of place cells between the two linear tracks (Pearson's correlation coefficient $r$ between tracks population vectors, $r = 0.11 \pm 0.13$), such that the majority of place

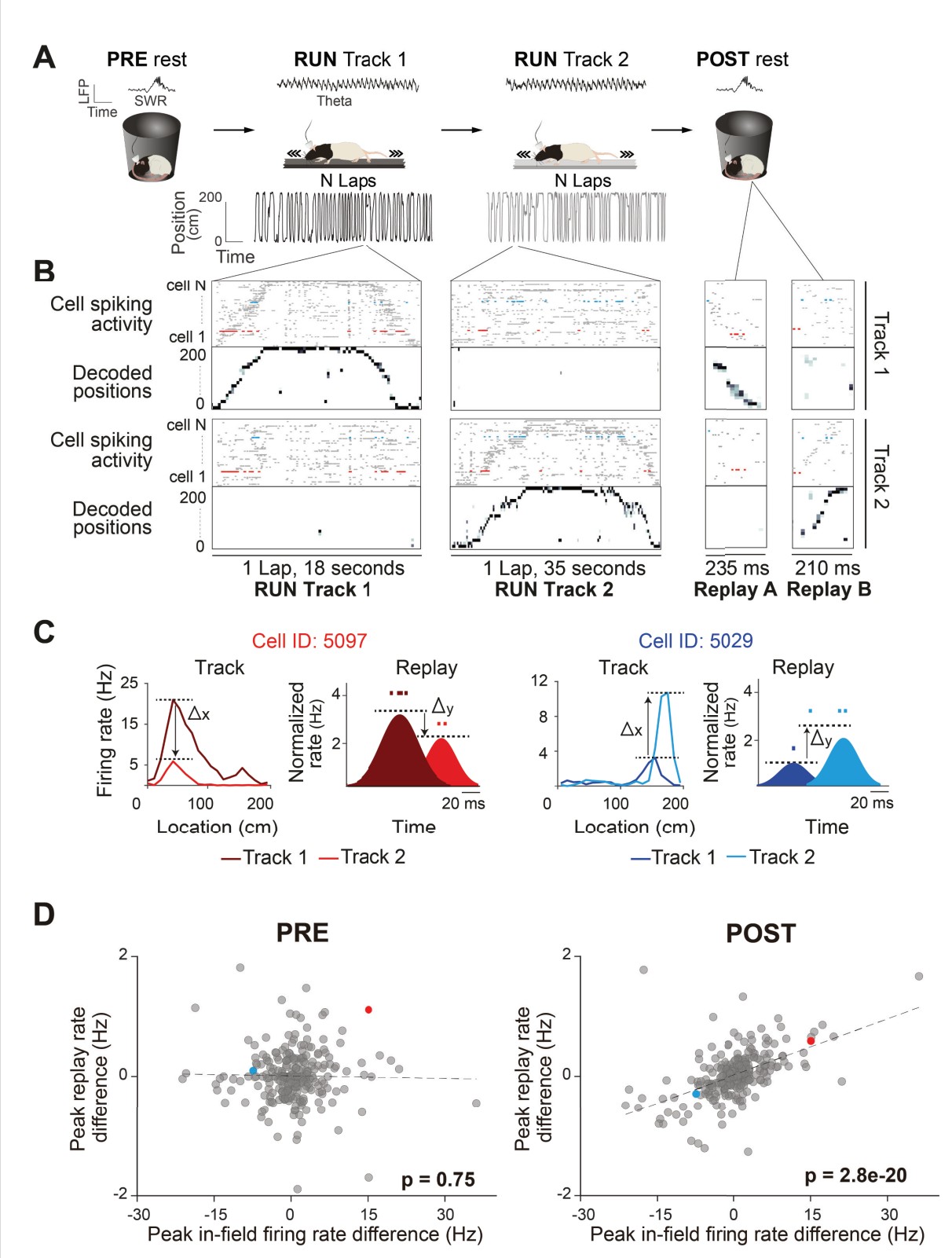

**Figure 1.** Rate coding in wake and rest. (**A**) Experimental design. Each recording session, the rats rested (PRE), ran back and forth on two different linear tracks (RUN), and then rested again (POST) (left to right). An example of LFP activity in each behavioral state is displayed (RUN-theta, PRE/POST-sharp wave ripples [SWR]). Rat schematic was adapted from SciDraw.io (https://doi.org/10.5281/zenodo.3926277, https://doi.org/10.5281/zenodo.3926237, https://creativecommons.org/licenses/by/4.0). (**B**) Raster plots of spiking activity of cells sorted by their peak firing rate on the track (top: Track 1, n =

*Figure 1 continued on next page*

*Figure 1 continued*

72 cells; bottom: Track 2, n = 57 cells), and the associated decoded positions for a single exemplar lap (first two columns) and replay event (last two columns), for each track (top: Track 1; bottom: Track 2). Because posterior probabilities are normalized across both tracks, decoded positions during behavior and replay can have substantially higher likelihood values on one track. Example from rat 3, session 2. (**C**) Two example neurons (highlighted in **B**) that changed their peak instantaneous firing rate (shaded areas, cartoon depictions of instantaneous firing rates) during two example replay events (displayed in **B**) in the same direction as their peak in-field firing rate difference between tracks (unshaded areas, actual ratemaps). The activity and properties of the example neurons are color coded throughout all the panels. (**D**) The peak in-field firing rate difference (Track 1–Track 2) significantly predicts the average peak instantaneous replay rate difference (Track 1–Track 2) in POST rest, but not PRE rest. Each data point is a neuron active on both tracks (PRE: n = 252 cells, POST: n = 272 cells).

The online version of this article includes the following figure supplement(s) for figure 1:

**Figure supplement 1.** Experimental setup, remapping, and field stability.

**Figure supplement 2.** Analysis pipeline for detection of significant replay events.

**Figure supplement 3.** Reinstatement of rate modulation during replay for each individual session.

**Figure supplement 4.** Significance of regression analysis using a stricter criterion of place cell activity.

**Figure supplement 5.** Significance of regression analysis with alternative place field and replay rate metrics, and stricter criteria for cell selection.

**Figure supplement 6.** Significance of regression analysis with alternative criteria for replay detection and cell selection.

fields shifted their location along the track and/or modulated their firing rate (i.e., changed to higher or lower peak in-field firing rate) (*Figure 1—figure supplement 1B*). Both the context (which linear track) and current position (location of the animal on the linear track) could be inferred from place cell activity with high accuracy using a naïve Bayes decoder (*Figure 1B*, *Figure 1—figure supplement 1C*). Track detection accuracy exceeded 80% during RUN periods on both tracks (all sessions: 93 ± 3%) with an average median error of 5 cm (all sessions: 6.26 ± 2.6 cm). A Population Vector Analysis showed high correlation between the ratemaps of the first and second half of running within each track (Pearson's correlation coefficient $r$ within track population vectors, $r = 0.81 \pm 0.11$), suggesting high place field stability over run. We also used Bayesian decoding to detect replay sequences of each linear track (*Figure 1B*), with statistically significant replay trajectories identified by comparing the weighted correlation score obtained from the posterior probabilities (normalized across both tracks) of each decoded event to three types of event shuffles (*Figure 1—figure supplement 2*; total replay events = 5396, PRE: Track 1 = 52 ± 33, Track 2 = 47 ± 23; POST: Track 1 = 145 ± 56, Track 2 = 149 ± 35; RUN: Track 1 = 68 ± 20, Track 2 = 79 ± 27; see 'Methods'). Replay events passing our criteria for significance comprised of 11.7% of PRE candidate replay events (989/8485) and 20.7% of POST candidate replay events (2964/14326).

First, we sought to investigate whether the contextually driven place fields' rate modulation between tracks is conserved during offline replay. We use the term 'contextually driven' to indicate a rate change occurring between the Track 1 and Track 2 contexts, albeit driven by environmental cues, track position, and/or behavioral differences in the rat. Using all place cells active and stable during the whole run on both tracks (*Figure 1—figure supplement 1B*; see 'Methods'), we compared the place fields' peak in-field firing rate differences between tracks (Track 1–Track 2) to the difference (Track 1–Track 2) in average peak instantaneous rate of those same cells during significant replay events (*Figure 1C*, *Figure 1—figure supplement 2*). For replay events during POST, the difference in peak in-field firing rate between tracks was predictive of the difference in average peak instantaneous rates during replay events from both tracks (*Figure 1D*; POST: B = 0.031, $F_{(1,270)} = 100$, $p<0.001$, $R^2 = 0.27$; *Figure 1—figure supplement 3* for individual sessions). In other words, if a place field had a higher firing rate on Track 2 (compared to Track 1), it also tended to have a higher mean firing rate across all Track 2 replay events (compared to Track 1 events) during POST. This was not the case during PRE (*Figure 1D*; PRE: B = −0.001, $F_{(1,250)} = 0.11$, $p=0.74$, $R^2 = 0.0004$). This suggests that any contextually driven rate modulation between two behavioral episodes is maintained during offline replay. This observation was consistent over a wide range of alterations in the criteria used for selecting neurons, place fields, and replay events, including (1) using a more selective criteria of place field activity (e.g., place fields with higher minimum or maximum peak firing rate or place cells with a single field) (*Figure 1—figure supplements 4 and 5D*), (2) alternative metrics for both replay firing rate (e.g., median replay rate and mean number of spikes per replay event) (*Figure 1—figure supplement 5B and C*) and place fields firing rate (e.g., place field's area under the curve) (*Figure 1—figure*

**Table 1.** Overview of behavior, number of recorded cells, and decoding accuracy for all sessions and rats.

| Session | Track | Number of laps | Time resting (min) | | Cells | | | Decoding quality (%) | |
|---|---|---|---|---|---|---|---|---|---|
| | | | PRE | POST | Cells used for replay analysis | Common to T1 and T2 (stable/non-stable) | Stable cells included in regression analysis | Classification accuracy | Local decoding accuracy |
| Rat 1, session 1 | 1 | 10 | 83 | 155 | 27 | 10 (9/1) | 9 | 86 | 58 |
| Rat 1, session 1 | 2 | 12 | 83 | 155 | 27 | 10 (9/1) | 9 | 93 | 66 |
| Rat 2, session 1 | 1 | 15 | 59 | 125 | 70 | 41 (29/12) | 29 | 97 | 86 |
| Rat 2, session 1 | 2 | 59 | 59 | 125 | 63 | 41 (29/12) | 29 | 93 | 89 |
| Rat 2, session 2 | 1 | 22 | 54 | 104 | 49 | 28 (17/11) | 17 | 92 | 82 |
| Rat 2, session 2 | 2 | 33 | 54 | 104 | 43 | 28 (17/11) | 17 | 90 | 79 |
| Rat 3, session 1 | 1 | 34 | 49 | 123 | 59 | 33 (25/8) | 25 | 94 | 85 |
| Rat 3, session 1 | 2 | 27 | 49 | 123 | 51 | 33 (25/8) | 25 | 94 | 80 |
| Rat 3, session 2 | 1 | 36 | 69 | 137 | 72 | 44 (33/11) | 33 | 94 | 85 |
| Rat 3, session 2 | 2 | 38 | 69 | 137 | 57 | 44 (33/11) | 33 | 97 | 86 |
| Rat 4, session 1 | 1 | 19 | 103 | 129 | 87 | 66 (52/14) | 52 | 94 | 80 |
| Rat 4, session 1 | 2 | 10 | 103 | 129 | 80 | 66 (52/14) | 52 | 94 | 85 |
| Rat 4, session 2 | 1 | 13 | 98 | 125 | 63 | 44 (35/9) | 35 | 88 | 78 |
| Rat 4, session 2 | 2 | 11 | 98 | 125 | 52 | 44 (35/9) | 35 | 97 | 82 |
| Rat 4, session 3 | 1 | 30 | 72 | 76 | 42 | 30 (20/10) | 20 | 91 | 81 |
| Rat 4, session 3 | 2 | 20 | 72 | 76 | 44 | 30 (20/10) | 20 | 97 | 82 |
| Rat 5, session 1 | 1 | 13 | 91 | 76 | 50 | 31 (20/11) | 20 | 97 | 85 |
| Rat 5, session 1 | 2 | 11 | 91 | 76 | 57 | 31 (20/11) | 20 | 90 | 89 |
| Rat 5, session 2 | 1 | 13 | 72 | 86 | 70 | 47 (32/15) | 32 | 94 | 88 |
| Rat 5, session 2 | 2 | 19 | 72 | 86 | 66 | 47 (32/15) | 32 | 94 | 87 |

*supplement 5A*), (3) using a more relaxed criteria for a significant replay event, including using only two two shuffles in replay detection, raising the replay detection p-value cutoff to 0.1, or not requiring a threshold for ripple power for candidate replay events (*Figure 1—figure supplement 6A, B, and D*), (4) using a stricter p-value cutoff (0.01) for replay detection (*Figure 1—figure supplement 6C*), and (5) using either all spatially tuned cells or only cells with a place field on a single track (rather than cells with stable place fields on both tracks) (*Figure 1—figure supplement 6E and F*). It is important to note that a significant regression would be expected when analyzing neurons with a place field only on one track, as they are expected to participate in replay events of this track, while being silent during the replay of the other track. As such, our regression analysis only analyzed place cells active on both tracks and stable across the whole run (*Figure 1—figure supplement 1B*; see 'Methods'). When we applied our analysis separately to either non-spatially selective cells (pyramidal neurons with firing

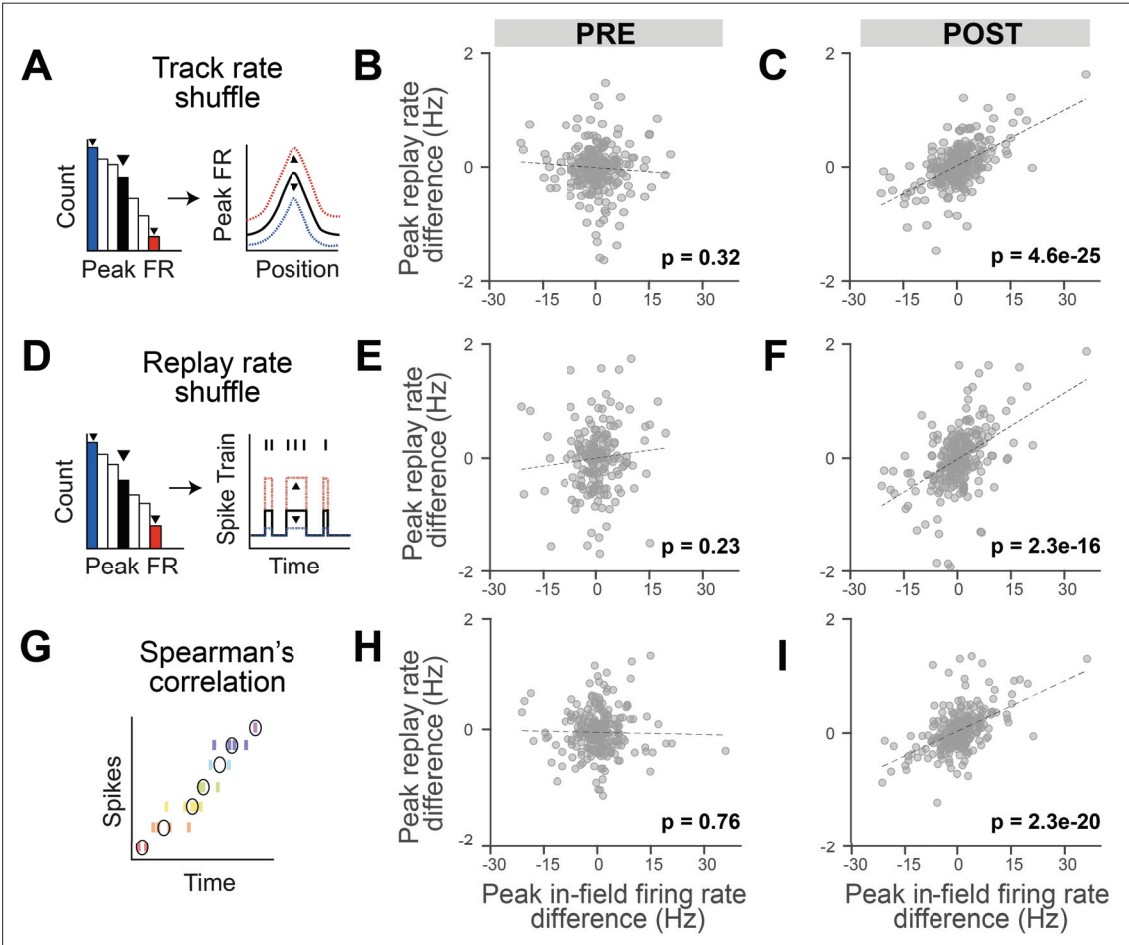

**Figure 2.** The reinstatement of rate modulation during rest replay is observed using replay detection methods insensitive to firing rate. (A, D, G) Three methods of replay detection that are insensitive to firing rate: (A) track rate shuffle (PRE: n = 252 cells, POST: n = 271 cells), (D) replay rate shuffle (PRE: n = 186 cells, POST: n = 261 cells), (G) rank-order correlation (Spearman), where black circles indicate median spike times (PRE: n = 253 cells, POST: n = 271 cells). (A, D) Example of the original rate (right plot, in black) randomly scaled up (right plot, in red) or down (right plot, in blue), after drawing a random value from the rate distribution (left plot). (B, E, H) Regression of peak in-field firing rate vs. peak replay rate difference (Track 1–Track 2) for PRE replay events with modified detection method: (B) track rate shuffle, (E) replay rate shuffle, and (H) Spearman's correlation. (C, F, I) Regression of peak in-field firing rate vs. peak replay rate difference (Track 1–Track 2) for POST replay events with modified detection method: (C) track rate shuffle, (F) replay rate shuffle, and (I) Spearman's correlation. The number of cells for each analysis varies as a consequence of different sets of replay events being respectively selected and therefore included in the regressions.

The online version of this article includes the following figure supplement(s) for figure 2:

**Figure supplement 1.** Between-track comparison of place field and replay properties.

**Figure supplement 2.** Recomputing the statistical significance of regression analysis relative to a shuffled distribution.

rates less than 1 spk/s on both tracks) or interneurons, we did not observe a significant regression (*Figure 1—figure supplement 6G and H*).

Replay detection methods generally detect the statistical significance of cells sequences based on templates derived from the behavioral episode, which are sensitive to the order of place field activation (during RUN and replay). However, because Bayesian decoding can also be influenced by firing rate, we next tested whether firing rate differences between tracks may bias the selection of the replay events used in the regression analysis. To remove the influence of firing rate in the replay event selection, we employed two independent types of shuffles before repeating the detection of replay events: (1) track rate shuffle, which randomizes the overall firing rate of place fields within a track (*Figure 2A–C*), and (2) replay rate shuffle, which randomizes the cell's firing rate during replay events (*Figure 2D–F*). For both shuffles, each place cell's firing rate was scaled based on a value randomly drawn from a distribution of firing rates obtained from the analyzed place cells on the track (for the track rate shuffle) or during the replay events (for the replay rate shuffle). These re-identified significant replay events were then used to repeat the regression analysis but using the original data (i.e., the original replay spike train and place fields, instead of the shuffled ones). For both shuffles, we observed a significant contextual rate modulation of replay, where track differences in the place fields' in-peak firing rate remained predictive of track differences in the average peak instantaneous rates during replay events in POST (*Figure 2C and F*; POST: track rate shuffle $p<0.001$, replay rate shuffle: $p<0.001$), but not PRE (*Figure 2B and E*; PRE: track rate shuffle $p=0.32$, replay rate shuffle: $p=0.23$). As an additional control, we detected replay events using a rank-order correlation method independent of Bayesian decoding (Spearman's correlation), which relies only on comparing the median spike times across the place cells of a replay event compared to the sequential order of place field activity along the track (*Figure 2G and I*). Repeating the regression analysis based on the track differences in the place fields' in-peak firing rate against the replay events detected using a Spearman's correlation still resulted in a significant contextual rate modulation of the replay events rate in POST, but not PRE (*Figure 2H and I*; PRE: $B = −0.001$, $F(1,251) = 0.09$, $p=0.76$, $R^2 = 0.0004$, POST: $B = 0.03$, $F(1,269) = 101$, $p<0.001$, $R^2 = 0.273$). Overall, these results indicate that the contextual rate modulation observed during replay is not a result of a bias within the replay detection analysis (namely, Bayesian decoding). Importantly, we did not observe any significant track differences in place field or replay properties (*Figure 2—figure supplement 1*), including the place fields' peak in-field firing rate (*Figure 2—figure supplement 1B*; two-sided two-sample Kolmogorov–Smirnov test, $p=0.59$) and the replay average peak instantaneous rate (*Figure 2—figure supplement 1B*; two-sided two-sample Kolmogorov–Smirnov test, $p=0.25$), and confirmed that our regression analysis was not influenced by any intrinsic biases between tracks by remeasuring the statistical significance relative to a shuffled distribution (*Figure 2—figure supplement 2*).

## Replay rate modulation emerges quickly with experience

The phenomenon of replay also occurs during awake quiescent states (e.g., while the animal pauses on the track) and has been proposed to serve different functional roles, ranging from planning to memory storage (*Diba and Buzsáki, 2007*; *Foster and Wilson, 2006*; *Gillespie et al., 2021*; *Pfeiffer and Foster, 2013*; *Xu et al., 2019*). We observed that during the behavioral episode on the tracks contextual rate modulation was expressed across local awake replay events (*Figure 3A*; RUN: $B = 0.05$, $F(1,268) = 186$, $p<0.001$, $R^2 = 0.41$). Rate modulation during replay was even more closely aligned with behavior during RUN compared to POST (*Figure 1D*; RUN: $F(1,268) = 186$, POST: $F(1,270) = 100$). Place cells during RUN already showed evidence of contextual rate modulation during awake local replay events within the first two laps of experience (*Figure 3B*; two laps: $B = 0.06$, $F(1,40) = 12.37$, $p=0.001$, $R^2 = 0.246$). Furthermore, place cells expressing rate modulation during RUN replay events were also likely to show a similar direction of contextual rate modulation (e.g., Track 1 > Track 2) during POST replay events (*Figure 3C*; $B = 0.83$, $F(1,268) = 173.9$, $p<0.001$, $R^2 = 0.39$). We did not observe a significant regression during PRE (*Figure 3—figure supplement 1A*; $B = 0.13$, $F(1,248) = 2.76$, $p=0.1$, $R^2 = 0.01$), including when our three alternative replay detection methods that controlled for rate biases were used (*Figure 3—figure supplement 1B–D*, $p>0.05$). Contextual rate modulation during replay was significantly expressed throughout the entire rest period during POST (at least 1 hr cumulatively, limited only by the maximum time period tested), with the strongest effect observed in the earliest portion of POST rest (*Figure 3D*).

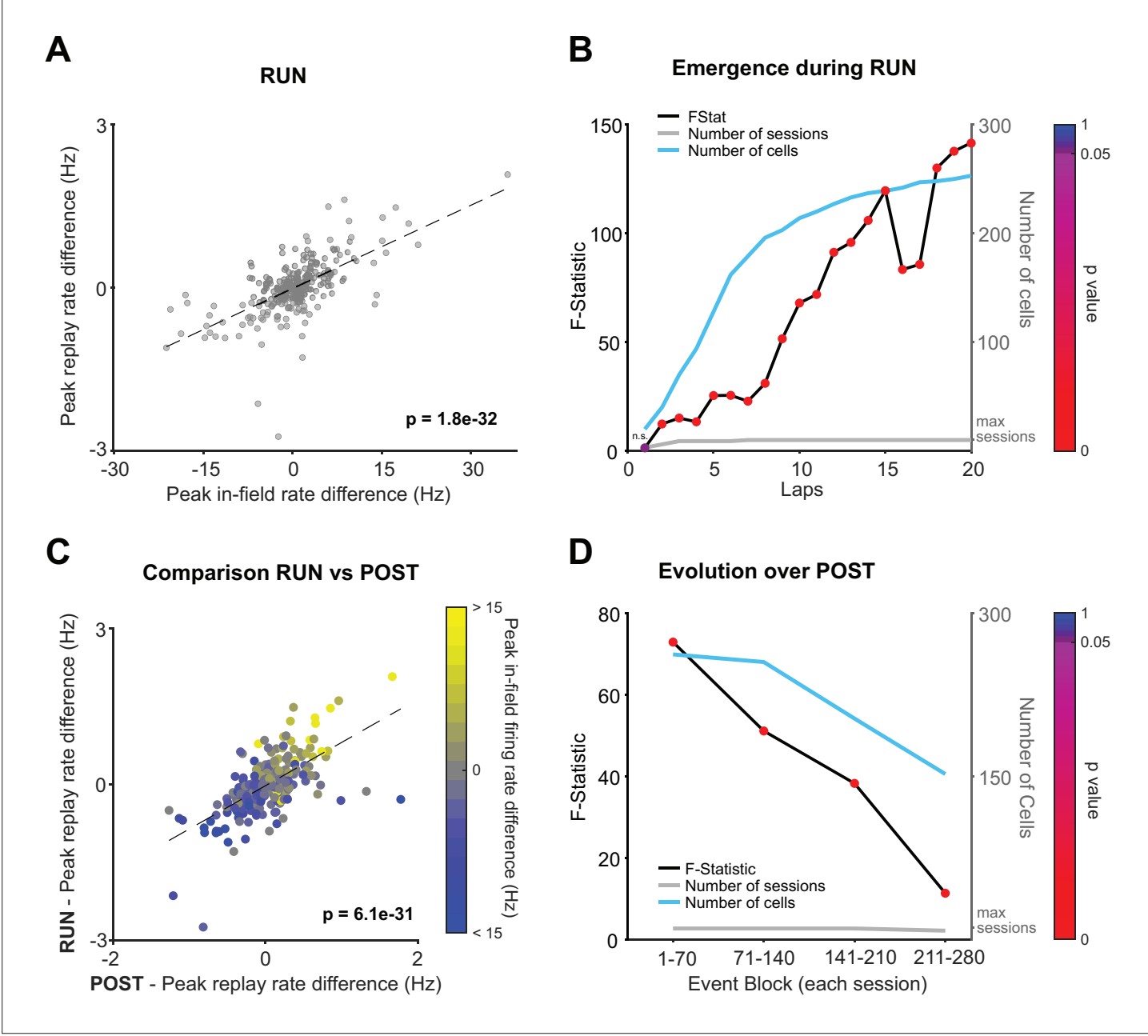

**Figure 3.** Rate modulation during replay is observed from the beginning of RUN and continues throughout the entire RUN and POST epochs. (**A**) Regression of peak in-field firing rates against peak replay rates (track differences) for all local RUN replay events. (**B**) Lap by lap emergence of rate modulation for RUN local replay events. F-statistic of regression – peak in-field firing rate vs. peak replay rate (track difference) – for replay events occurring both prior to and including the specified lap. The p-value of each regression is indicated by color. The number of cells and number of data sessions contributing in each regression are indicated with the cyan and gray line, respectively. (**C**) Peak replay rate difference for replay events (RUN vs. POST), color coded by peak in-field rate differences between tracks. (**D**) Temporal dynamics of rate modulation during POST. The regression's F-statistic (y-axis) and the corresponding p-value (marker color), analyzed in non-overlapping blocks of 70 consecutive replay events (per session). The number of cells and number of data sessions contributing in each regression are indicated with the cyan and gray line, respectively.

The online version of this article includes the following figure supplement(s) for figure 3:

**Figure supplement 1.** Between track differences in replay rate during PRE is not predictive of replay rate differences observed during RUN local replay events.

## Replay rate representations can be used to discriminate between contexts

Our data suggest that both place (i.e., sequences) and rate information is embedded within hippocampal replay events. If this is the case, then the spatial context (i.e., track identity) associated with a replay event should be identifiable using either the place or rate representations alone. To address this question, we modified the current Bayesian framework such that it would rely on either or both place and rate information (*Figure 4A*). For each decoded replay event, we measured the probabilistic bias towards one of the two tracks using the z-scored log odds (*Carey et al., 2019*), $\left( \log \frac{\sum prob(Track_1)}{\sum prob(Track_2)} \right)$. Thus, for a given replay event, this method summed the decoded posterior probability for each track, quantified the logarithmic ratio between the two tracks, and then z-scored this ratio relative to a shuffled distribution (ratemap track label shuffle, *Figure 4—figure supplement 1*; see 'Methods'). For example, a higher positive z-scored log odds would indicate a greater probabilistic bias towards Track 1. Therefore, if the content of a replay event was detected for Track 1 (based on a statistically significant sequence), the corresponding z-scored log odds would reflect how well a place and/or rate representation could be used to correctly classify this replay event as Track 1 (rather than Track 2) (*Figure 4B*).

In particular, we used two measurements to assess the replay track identity classification performance when using either the rate and/or place representation (*Figure 4—figure supplement 1*; see 'Methods'): (1) the log odds difference between two tracks compared to a replay track identity shuffled distribution and (2) the binary discriminability of a replay event's track identity using receiver-operative characteristic (ROC) curves, in which the area under the ROC curve (AUC) of 0.5 indicates chance discriminability (and an AUC of 1 indicates perfect discriminability). For both measurements, only replay events with a statistically significant replay sequence were used. When both place and rate information was unaltered, the z-scored log odds difference and the track discriminability were highest for RUN (Δ(z-scored log odds) = 2.61 ± 0.25, p<0.001, AUC = 0.97), significant but marginally lower for POST (Δ(z-scored log odds) = 1.19 ± 0.18, p<0.001, AUC = 0.77), but indistinguishable from the null distribution and at chance levels for PRE (Δ(z-scored log odds) = 0.05 ± 0.10, p=0.24, AUC = 0.53) (*Figure 4C and D*, *Table 2*). Because the number of cells common to both tracks varies between sessions, and affects classification accuracy, we compared the number of cells with the mean log odds difference between tracks for each session (*Figure 4—figure supplement 2*). A higher number of cells was significantly predictive of a higher log odds difference for POST (B = 0.0338, F(1,8) = 7.91, p=0.02, $R^2$ = 0.50) and RUN (B = 0.055, F(1,8) = 18.93, p=0.002, $R^2$ = 0.70), but not PRE (B = 0.011, F(1,8) = 1.89, p=0.21, $R^2$ = 0.19). An example of this is the presence of a session with lower log odds difference value during RUN in *Figure 4C* (n = 9 cells).

We then proceeded to selectively remove either the firing rate or place information available to the Bayesian decoder and assess its ability to determine the track being replayed. First, we removed the firing rate information by both setting each place field's in-field peak firing rate to its average across both tracks and rescaling each cell's replay spike trains to its average firing rate across all replay events (rate fixed manipulation). We found that the remaining information, arising most likely from ensemble co-activity – that is, which cells were active in a decoding bin – was sufficient to correctly classify replay events to each track, although both the z-scored log odds difference and the AUC were slightly lower than when both place and firing rate information were available (*Figure 5A and E–G*; AUC difference <0.1 compared to the original for all task periods, *Table 3A*).

Next, we selectively removed the sequential spatial and temporal relationship between neighboring place fields without altering rate information by expanding the size of both the position and time bins during the decoding: a single position bin spanning the entire track and a single time bin spanning the entire replay event (place removed manipulation, *Figure 5B and E–G*). This manipulation still led to a significant log odds difference towards the correct track and a high track discriminability for RUN (Δ(z-scored log odds) = 1.17 ± 0.20, p<0.001, AUC = 0.80), POST (Δ(z-scored log odds) = 0.52 ± 0.12, p<0.001, AUC = 0.65), but not PRE (Δ(z-scored log odds) = –0.04 ± 0.05, p=0.43, AUC = 0.53), supporting the finding that rate modulation is reinstated during replay and is experience-dependent. Finally, as a negative control for both manipulations, where rate and place representations were disrupted, each remaining non-manipulated representation (place or rate) was then randomized (rate fixed place randomized manipulation and place removed rate randomized manipulation,

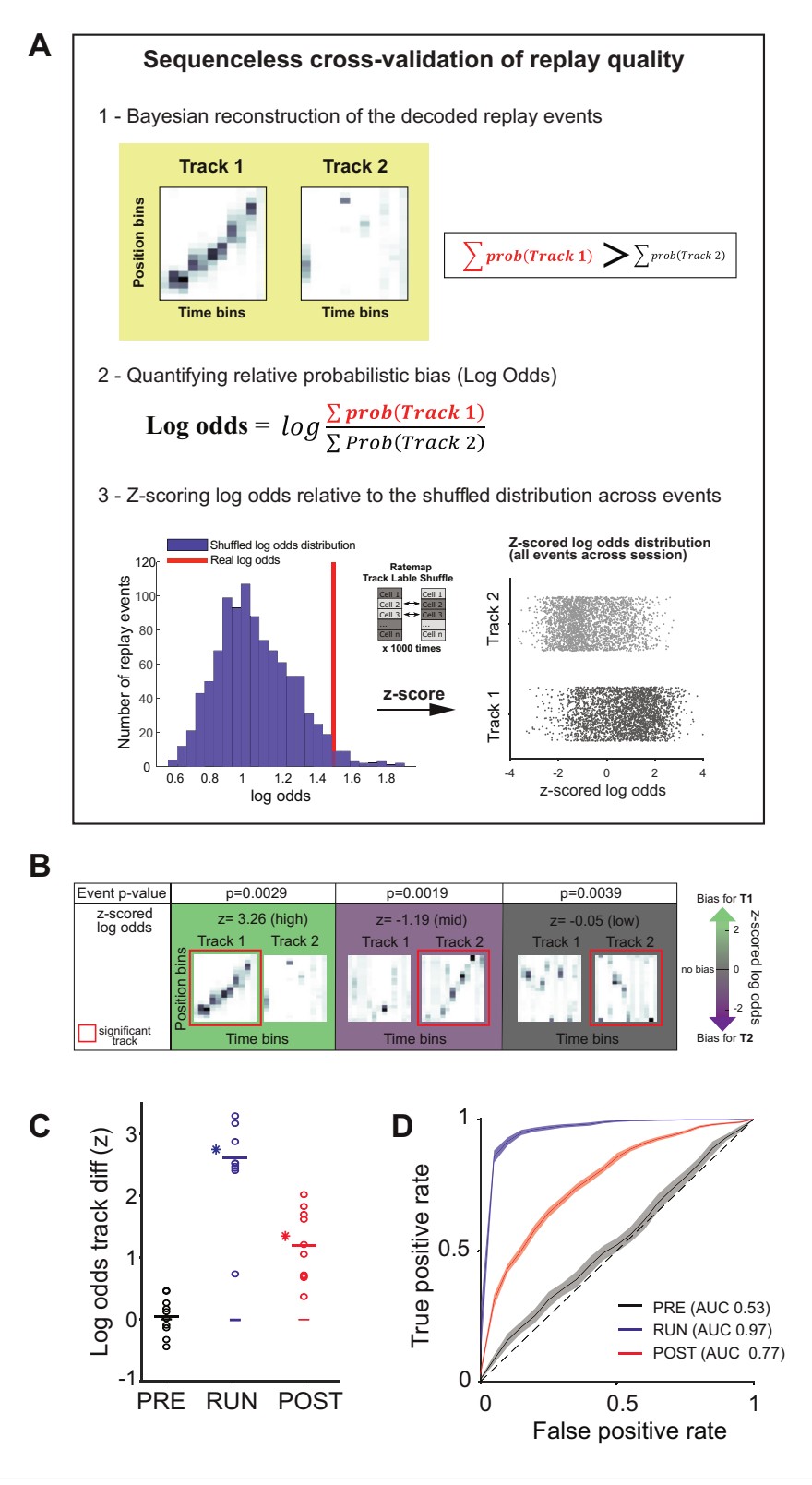

**Figure 4.** Sequenceless decoding to cross-validate replay events. (**A**) Description of a *sequenceless* decoding based method for cross-validating replay events originally detected based on *sequenceness*. (1) Events are decoded, and the posterior probabilities summed across each track, (2) the log odds ratio is calculated, and (3) this ratio is z-scored using a log odds distribution, based on place fields with shuffled track identity. (**B**) Example

*Figure 4 continued on next page*

*Figure 4 continued*

of three replay events scored as significant based on their sequenceness (i.e., order of cell activation) (top row 'Event p-value'), while presenting different posterior probability distributions across the two tracks (bottom row 'z-scored log odds'). From left to right, examples of replay events with high, medium, and low z-scored log odds, respectively. (**C**) The decoding performance for context (Track 1 or Track 2) using the mean z-scored log odds difference between tracks for replay during PRE (black), RUN (blue), and POST (red). Each data point is the mean of an individual session, while the horizontal line indicates the mean across all sessions (*p<0.05, one-tailed rank-sum test). The filled box indicates the SD from the replay track identity shuffles. (**D**) Receiver-operative characteristic (ROC) curves of Track 1/Track 2 binary discrimination for replay events during PRE, RUN, and POST. The dashed line along the diagonal indicates chance performance (area under the ROC curve [AUC] = 0.5). The shaded region indicates the bootstrapped SE of the ROC curves.

The online version of this article includes the following figure supplement(s) for figure 4:

**Figure supplement 1.** Schematic of modified Bayesian decoding framework.

**Figure supplement 2.** The relationship between the number of stable place cells that participated in replay and the mean log odds track difference during PRE (black), RUN (blue), and POST (red).

respectively). Removing both place and rate representations resulted in chance-level discriminability for both conditions (*Figure 5C–G*, *Table 3B*).

## Discussion

Here, we have shown that contextually driven place and rate representations used by the hippocampus during behavior are reinstated during replay, with either representation being sufficient to successfully discriminate which context is being replayed. These results support a model in which the hippocampus can encode more than the order of previously experienced place cell firing, with firing rates actively changing for each replay event to reflect contextual changes that have previously occurred between different behavioral episodes. This phenomenon is experience-dependent as rate modulation within replay was only observed during and after a behavioral episode, but not prior to it. Importantly, our paradigm allows us to uniquely demonstrate that rate modulation during local awake replay emerges rapidly with experience of novel environments.

The concept that a place cell's firing rate during a reactivation event is influenced by its firing rate during behavior has been put forward by several previous studies (*Pavlides and Winson, 1989*; *Huxter et al., 2003*; *Ravassard et al., 2013*; *Savelli and Knierim, 2019*; *Schwindel et al., 2016*; *Takahashi, 2015*; *Farooq et al., 2019*). However, one challenge in interpreting these studies is that reactivation methods typically use firing rate in their detection, in turn potentially biasing their selection to events with rate modulation. To avoid the potential for such circular reasoning, and more rigorously demonstrate the reinstatement of rate modulation, here we incorporate four key advances over previous studies. First is our analysis of replay rather than reactivation, which uses the sequential pattern of firing across place cells for detection, in turn providing significantly more statistical power than using a sequenceless-based reactivation approach. Our methods allowed us to still detect events when firing rate information is removed, which we accomplished in three different ways (*Figure 2*). Second is our use of two distinct behavioral tracks within a recording session (*Karlsson and Frank, 2009*; *Gupta et al., 2010*), allowing us to compute the relative change in firing rates between tracks (both for behavior and replay). This avoids the problem of slow variations in neuronal excitability potentially confounding the measurement of correlation between behavior and sleep, when the analysis is limited to the absolute firing rates within a single context. It is worth noting that this study has focused on global remapping as testing for reinstated rate modulation within a rate remapping paradigm lacks the ability to cross-reference rate changes using rate-independent replay content decoding. Third, previous studies using multiple tracks did not remove neurons that only fired in one context from their analysis, which can artificially boost the detection of rate modulation. This is because the replay of a track activates place fields from that same track, but these same place cells should be silent when another track is replaying (if no place field exists on this second track). For this reason, our analysis of rate modulation focused exclusively on neurons with stable place fields on both tracks, and similar results were obtained using higher thresholds for what qualified as a place field (*Figure 1—figure supplement 4*). Finally, our approach provides a new method to cross-validate

**Table 2.** Summary statistics of the z-scored log odds difference per session and across sessions for PRE, RUN, and POST.

| | Session | Original | | | Rate fixed | | | Place removed | | | Rate fixed, place randomized | | | Place removed, rate randomized | | |
|---|---|---|---|---|---|---|---|---|---|---|---|---|---|---|---|---|
| | | Mean | SE | p-Value | Mean | SE | p-Value | Mean | SE | p-Value | Mean | SE | p-Value | Mean | SE | p-Value |
| PRE | Rat1 Session1 | −0.09 | | | 0.32 | | | 0.02 | | | −0.03 | | | −0.30 | | |
| | Rat2 Session1 | 0.27 | | | 0.38 | | | 0.02 | | | 0.04 | | | 0.00 | | |
| | Rat Session2 | −0.32 | | | −0.51 | | | −0.34 | | | 0.31 | | | 0.03 | | |
| | Rat3 Session1 | 0.13 | | | −0.21 | | | 0.06 | | | −0.37 | | | 0.05 | | |
| | Rat3 Session2 | 0.47 | | | 0.65 | | | 0.00 | | | −0.05 | | | 0.17 | | |
| | Rat4 Session1 | 0.17 | | | 0.22 | | | 0.06 | | | 0.12 | | | −0.08 | | |
| | Rat4 Session2 | 0.02 | | | 0.27 | | | −0.05 | | | 0.06 | | | 0.23 | | |
| | Rat4 Session3 | −0.13 | | | −0.01 | | | −0.04 | | | −0.19 | | | −0.18 | | |
| | Rat5 Session1 | −0.44 | | | −0.17 | | | −0.35 | | | 0.23 | | | −0.04 | | |
| | Rat5 Session2 | 0.46 | | | 0.26 | | | 0.18 | | | 0.13 | | | −0.11 | | |
| | Mean | 0.05 | 0.10 | 0.24 | 0.12 | 0.11 | 0.21 | −0.04 | 0.05 | 0.43 | 0.03 | 0.06 | 0.24 | −0.02 | 0.05 | 0.69 |

*Table 2 continued on next page*

*Table 2 continued*

| | | Original | | Rate fixed | | Place removed | | Rate fixed, place randomized | Place removed, rate randomized |
|---|---|---|---|---|---|---|---|---|---|
| RUN | Rat1 Session1 | 0.74 | | −0.19 | | −0.11 | | −0.34 | 0.05 |
| | Rat2 Session1 | 3.29 | | 2.36 | | 1.47 | | −0.05 | −0.07 |
| | Rat2 Session2 | 2.53 | | 2.31 | | 1.22 | | −0.29 | −0.21 |
| | Rat3 Session1 | 2.48 | | 2.42 | | 0.35 | | −0.35 | −0.05 |
| | Rat3 Session2 | 3.17 | | 2.23 | | 1.52 | | −0.18 | 0.06 |
| | Rat4 Session1 | 3.67 | | 3.00 | | 2.04 | | 0.18 | 0.23 |
| | Rat4 Session2 | 2.88 | | 2.02 | | 1.40 | | −0.31 | −0.18 |
| | Rat4 Session3 | 2.45 | | 2.10 | | 1.11 | | 0.03 | −0.32 |
| | Rat5 Session1 | 2.41 | | 1.89 | | 1.07 | | 0.17 | −0.25 |
| | Rat5 Session2 | 2.53 | | 1.80 | | 1.67 | | 0.10 | −0.14 |
| | Mean | 2.61 | 0.25 | 1.99 | 0.26 | 1.17 | 0.20 | −0.11 | −0.09 |
| | | 9.13 × 10⁻⁵ | | 1.41 × 10⁻³ | | 1.41 × 10⁻³ | | 0.07 | 0.05 |
| | | | | | | | | 0.79 | 0.94 |

*Table 2 continued on next page*

Table 2 continued

| | Original | Rate fixed | Place removed | Rate fixed, place randomized | Place removed, rate randomized |
|---|---|---|---|---|---|
| **POST** Rat1 Session1 | 0.37 | 0.06 | −0.06 | −0.05 | −0.09 |
| Rat2 Session1 | 1.70 | 1.11 | 0.59 | 0.21 | −0.03 |
| Rat2 Session2 | 1.62 | 1.40 | 0.80 | −0.36 | 0.08 |
| Rat3 Session1 | 0.70 | 0.78 | 0.18 | −0.13 | −0.07 |
| Rat3 Session2 | 1.83 | 1.57 | 0.87 | 0.05 | −0.13 |
| Rat4 Session1 | 2.02 | 1.51 | 1.12 | 0.09 | −0.01 |
| Rat4 Session2 | 1.06 | 0.80 | 0.49 | −0.04 | 0.07 |
| Rat4 Session3 | 0.72 | 0.34 | 0.26 | 0.17 | −0.09 |
| Rat5 Session1 | 0.68 | 0.42 | 0.21 | −0.05 | 0.10 |
| Rat5 Session2 | 1.21 | 0.97 | 0.73 | −0.05 | 0.01 |
| Mean | 1.19, 0.18, $9.13 \times 10^{-5}$ | 0.90, 0.16, $9.13 \times 10^{-5}$ | 0.52, 0.12, $1.41 \times 10^{-3}$ | −0.02, 0.05, 0.79 | −0.02, 0.03, 0.79 |

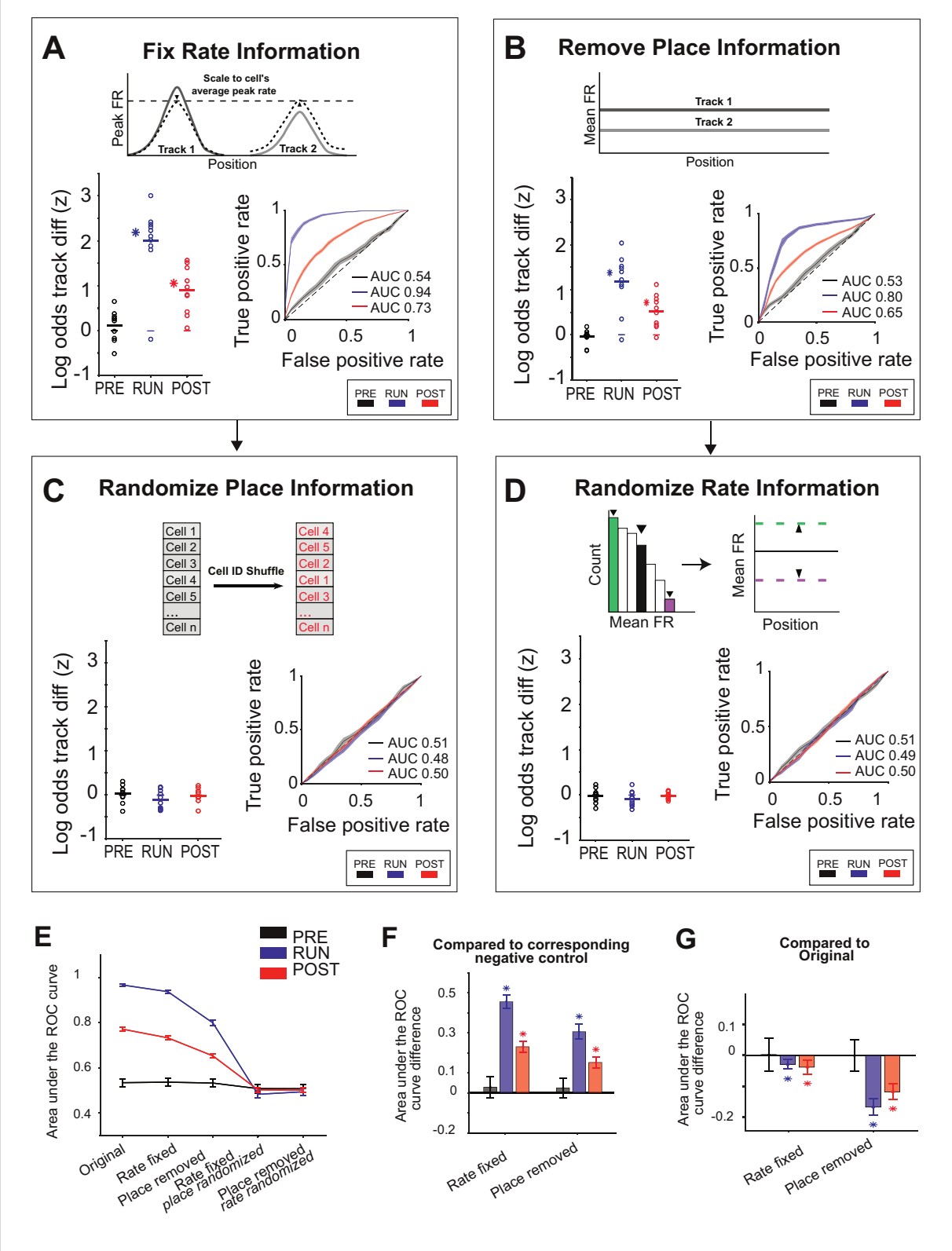

**Figure 5.** Either place or rate representations can be used to classify the replayed context. (**A–D**) The decoding performance for context (Track 1 or Track 2) for each manipulation, restricting the information (rate and/or place) available to the decoder. (Main plots) Mean z-scored log odds difference between tracks for replay during PRE (black), RUN (blue), and POST (red). Each data point is the mean of an individual session, while the horizontal line indicates the mean across all sessions (*p<0.05, one-tailed rank-sum test). The filled box indicates the SD from the replay track identity shuffles. (Insets)

*Figure 5 continued on next page*

*Figure 5 continued*

Receiver-operative characteristic (ROC) curves of Track 1/Track 2 binary discrimination for replay events during PRE, RUN, and POST. The dashed line along the diagonal indicates chance performance (area under the ROC curve [AUC] = 0.5). The shaded region indicates the bootstrapped SE of the ROC curves. (**A**) Rate fixed manipulation (only place information available), (**B**) place removed manipulation (only rate information available), (**C**) rate fixed manipulation with place information additionally randomized (no rate or place information available), and (**D**) place removed manipulation with rate information additionally randomized (no rate or place information available). (**E**) Summary of mean and SEM AUC under each manipulation condition, for PRE, RUN, and POST. (**F**) Mean AUC difference between each manipulation and the corresponding negative control. (**G**) Mean AUC difference between original and rate fixed and place removed manipulation. For (**F, G**), asterisks (*) indicate where the 95% confidence interval of the bootstrapped AUC difference distribution does not overlap with 0 (*Table 3A and B*). Similar results were obtained using an alternative approach (DeLong method) for measuring significance (*Table 3C and D*; see 'Methods').

replay events by comparing whether the track with the significant sequence matches the track with the higher likelihood of occupancy (log odd analysis). This allows us to verify that both rate and place information present in replay events can independently be used downstream to identify the context being replayed.

What advantage does the reinstatement of firing rate during replay provide for memory consolidation? When minor modifications are made to an environment (e.g., changing the wall color), place cells can change their in-field firing rates without altering the location of their place fields, a phenomenon referred to as rate remapping (*Leutgeb et al., 2005*). As such, replay sequences would not be able to discriminate between behavioral episodes from before and after the contextual change. Our results present a possible solution to this problem as two marginally similar contexts are still potentially separable due to their rate modulation differences. It is also important to consider that computational models of replay have commonly used modified synfire chains, sequentially connected neurons that are often embedded within distinct recurrent networks (*Abeles, 1982*; *Chenkov et al., 2017*; *Diesmann et al., 1999*). While these models aim to replicate the sequential nature of replay, it remains to be seen whether they can also recreate experience-dependent rate modulation or whether additional mechanisms may need to be considered. Experimental evidence indicates that neocortical activity preceding a hippocampal replay event plays a key role in determining its content (*Bendor and Wilson, 2012*; *Lewis and Bendor, 2019*; *Rothschild et al., 2017*). As such, if cortical inputs to the hippocampus during behavior underlie rate modulation (*Renno-Costa and Tort, 2017*), a reinstatement of these same inputs prior to replay may be necessary for the subsequent rate modulation of hippocampal place fields during offline states.

## Methods

### Animals

Five male Lister-hooded rats (350–450 g) were implanted with a microdrive with 24 independently moveable tetrodes. Prior to surgery, rats were kept at 90% of their free-feeding weight and housed in pairs on a 12 hr light/dark cycle, with 1 hr of simulated dusk/dawn. All experimental procedures and postoperative care were approved and carried out in accordance with the UK Home Office, subject to the restrictions and provisions contained within the Animal (Scientific Procedures) Act of 1986.

### Surgery

Animals were deeply anesthetized under isoflurane anesthesia (1.5–3% at 2 L/min) and implanted with a custom-made microdrive array carrying 24 independently moveable tetrodes (modified from microdrive first published by *Davidson et al., 2009*). Each tetrode consisted of a twisted bundle of four tungsten microwires (12 µm diameter, Tungsten 99.95% CS, California Fine Wire), gold-plated to reduce impedance to <200 kΩ. Three rats were implanted with a dual-hippocampal microdrive targeting both dorsal hippocampal CA1 areas (AP: –3.48 mm; ML: ±2.4 mm from Bregma), each output carrying 12 tetrodes. The two remaining rats were implanted with a microdrive targeting the right dorsal hippocampal CA1 area (AP: 3.72 mm, ML: 2.5 mm from Bregma) and the left primary visual cortex (AP: –5.76 mm; ML: –3.8 mm from Bregma), using 16 and 8 tetrodes, respectively. After surgery, animals were housed individually and allowed to recover with food and water ad libitum for a week before returning to being kept at 90% of their free-feeding weight.

**Table 3.** Summary statistics for the binary discriminability of a replay event's track identity using receiver-operative characteristic (ROC) curves under the original and manipulated conditions. (A) Summary statistics of the 95% confidence interval for the area under the receiver-operative characteristic (AUC) curve difference between the original and the manipulated conditions. (B) The manipulated conditions and their respective negative controls. (C) Summary statistics of the DeLong test comparing the ROC curves between the original and the manipulated condition. (D) Summary statistics of the DeLong Test comparing the ROC curves between the manipulated conditions and their respective negative controls.

| A | | Vs. original 95% confidence interval | | |
|---|---|---|---|---|
| | | PRE | RUN | POST |
| | Mean AUC Diff | 0.002 | –0.03 | –0.04 |
| | SEM AUC Diff | 0.03 | 0.01 | 0.01 |
| | Lower CI | –0.05 | **–0.05** | **–0.06** |
| Original vs. rate fixed | Upper CI | 0.06 | **–0.02** | **–0.02** |
| | Mean AUC Diff | –0.002 | –0.17 | –0.12 |
| | SEM AUC Diff | 0.03 | 0.01 | 0.01 |
| | Lower CI | –0.06 | **–0.20** | **–0.14** |
| Original vs. place removed | Upper CI | 0.05 | **–0.14** | **–0.09** |

| B | | Vs. negative control 95% confidence interval | | |
|---|---|---|---|---|
| | | PRE | RUN | POST |
| | Mean AUC Diff | 0.03 | 0.46 | 0.23 |
| | SEM AUC Diff | 0.03 | 0.02 | 0.01 |
| | Lower CI | –0.03 | **0.42** | **0.20** |
| Rate fixed vs. rate fixed + place randomized | Upper CI | 0.08 | **0.49** | **0.26** |
| | Mean AUC Diff | 0.03 | 0.31 | 0.15 |
| | SEM AUC Diff | 0.03 | 0.02 | 0.01 |
| | Lower CI | –0.02 | **0.27** | **0.13** |
| Place removed vs. place removed + rate randomized | Upper CI | 0.08 | **0.34** | **0.18** |

| C | DeLong test vs. original (significance = 0.05) | |
|---|---|---|
| | Original vs. rate fixed | Original vs. place removed |
| PRE | 0.89 | 0.95 |
| RUN | $1.84 \times 10^{-6}$ | $6.48 \times 10^{-52}$ |
| POST | $1.37 \times 10^{-5}$ | $3.42 \times 10^{-35}$ |

| D | DeLong test vs. negative control (significance = 0.05) | |
|---|---|---|
| | Rate fixed vs. rate fixed + place randomized | Place removed vs. place removed + rate randomized |
| PRE | 0.24 | 0.31 |
| RUN | $1.94 \times 10^{-163}$ | $1,00 \times 10^{-55}$ |
| POST | $3.00 \times 10^{-64}$ | $1.82 \times 10^{-28}$ |

Bold values statistically significant from zero.

## Experimental design

A given recording session started with a 1 hr rest period in which the rats were placed in a quiet, remote location (rest pot), to which they had been previously habituated. The rest pot consisted of a black circular enclosure of 20 cm of diameter, surrounded by a 50 cm tall black plastic sheet that isolated them from the surroundings. The animals went through one of the two following protocols:

1. Following the rest period, the rats were exposed to two novel 2 m linear tracks in which they were allowed to run back and forth for 15 min, except for one session in which the animal ran for 30 min in the second track. (Rat 4 session 3; Rat 2 and Rat 3 all sessions.)
2. Following the rest period, the rats were exposed to three novel 2 m linear tracks in which they were allowed to run back and forth for 15 min. Data from the first track has been removed for this study to ensure consistency between protocols, such as temporal proximity to final rest session, and for all analyses. (Rat 1 session 1; Rat 4 sessions 1 + 2, and Rat 5 all sessions.)

Prior to the start of recordings, rats were trained for approximately 2 days, 30 min each, to run back and forth on a linear track with reward delivered at each end. Training occurred in a different room and track from the one used during the recordings. Liquid reward was dispensed at each end of the track (0.1 mL chocolate-flavored soy milk) to encourage the animals to traverse the entirety of the track. In all except one session (Rat 2 session 2), the exposures to the two tracks were separated by a 10 min rest period in the rest pot. The recording session finished with a final 2 hr rest period inside the rest pot.

The linear tracks used in every experiment were designed to be novel to the animal. To accomplish this, the shape of the tracks was changed between recording sessions and their surfaces covered with different textured fabrics. In each session, the room was surrounded by black curtains with different high-contrast visual cues. The tracks were separated using view-obstructing dividers.

## Spike detection and unit isolation

Spiking data was sorted using the semi-automatic clustering software KlustaKwik 2.0 (K.Harris, http://klustakwik.sourceforge.net/) and then manually curated with either Phy-GUI (https://github.com/kwikteam/phy; *Rossant et al., 2022*; *Rossant et al., 2016*) or Klustaviewa (https://github.com/klusta-team/klustaviewa; *Rossant et al., 2017*; *Rossant et al., 2016*). Putative single units were discriminated based on the spike waveform, a clean inter-spike interval, and their stability across the recording session. The rest of the clustered activity was classified in either multi-unit activity (MUA) or noise.

## LFP analysis

The power spectral density (PSD) of the LFP was calculated using Welch's method (*pwelch*, MATLAB) to identify the channels with higher power for theta (4–12 Hz) and ripple (125–300 Hz) oscillations, as well as the channel with the largest difference in normalized theta to ripple power. The LFP of selected channels was next downsampled from 30 kHz to 1 kHz and band-passed filtered (MATLAB command *filtfilt*). The instantaneous phases were estimated using Hilbert transform.

## Criteria for place cell selection

Putative principal cells were identified by selecting units with a half width half max (HWHM) larger than 500 μs and mean firing rate <5 Hz across the entire recording session. For place cell classification, spike trains were speed-filtered to only include the spiking activity between 4 cm/s and 50 cm/s. A principal cell was classified as a place cell if it had a minimum peak firing rate that was >1 Hz in its unsmoothed ratemap. Furthermore, place cells were required to have stable place fields, defined as a minimum peak firing rate >1 Hz for both the first half and second half of the RUN session. Only place cells satisfying these two criteria for each linear track independently were included in the regression analysis (*Figures 1–3*) and log odds analysis (*Figures 4 and 5*).

To generate firing ratemaps (the spike histogram divided by the total dwell time at each position bin), the position data was discretized in 2 cm bins for visualization and plotting, and 10 cm bins for Bayesian decoding. Only raw (unsmoothed) ratemaps were used for all Bayesian decoding analyses.

## Population vector analysis

A Population Vector Analysis was used to assess the ratemap stability for each track as well as the degree of between track remapping (*Leutgeb et al., 2005*). For each session, the linearized ratemaps of all recorded pyramidal cells with a firing rate >1 Hz were stacked into a 20 position bins × N cells matrix for each track. To assess remapping between tracks, the ratemaps computed from the entire track experience were compared, while to assess ratemap stability within a track, a ratemap was computed for both the first and second half of the behavioral episode. The population vector of rates at each position bin was then correlated with its counterpart vector: either on the other track or during the other half of the experience. The mean Pearson's correlation value was first averaged across position bins and then across sessions.

## Bayesian decoding

A naïve Bayesian decoding algorithm was applied to reconstruct the animal's spatial trajectory during both behavior and replay events from CA1 hippocampal spiking activity (*Zhang et al., 1998*):

$$P\left(x|n\right) = C\left(\prod_{i=1}^{N} f_i\left(x\right)^{n_i}\right) exp\left(-\tau \sum_{i=1}^{N} f_i\left(x\right)\right)$$

where P(x|n) is the probability of the animal being at a specific position given the observed spiking activity, C is a normalization constant, x is the subject's position, $f_i(x)$ is the firing rate of the ith place field at a given location x, and n is the number of spikes in the time window $\tau$. The normalization constant was redefined as the summed posterior probabilities across both tracks. Since we are using a naïve decoder, the prior P(x) is uniform across all position and therefore set to 1. The maximum probability of the subject's position was decoded using 250 ms and 20 ms time bin during behavior and replay, respectively. All place cells with a place field on at least one of the two tracks were used for decoding.

The decoding error was defined as the differences between the real location of the animal and the estimated position with maximum likelihood.

## Candidate replay event detection

Candidate replay events were detected based on MUA. MUA was first smoothed with a Gaussian kernel (sigma = 5 ms) and binned into 1 ms steps. Only MUA bursts with a maximum duration of 300 ms and z-scored activity over 3 were selected.

Candidate replay events passing this threshold were next speed-filtered (the animal's speed was required to be less than 5 cm/s) and discarded if their sequence had less than five different units active or if their duration was below 100 ms or over 750 ms (thus, candidate events had at least five consecutive 20ms bins). Events detected within 50 ms were combined. Additionally, the ripple-band filtered LFP signal was smoothed with a 0.1 s moving average filter and a threshold over 3 was set for the z-scored ripple power, and used as an additional criterion for candidate replay events. In an effort to optimize detection of replay events and avoid a minority of events that were discarded due to noisy probability decoding at the beginning or the end of the event, candidate replay events were split in half. To do so, the minimum MUA activity in the middle third of the candidate replay event was used to determine a natural midpoint to split the event in two segments. Both segments were decoded and tested for significance independently following the same procedure as the 'intact' candidate replay events (i.e., same criteria including minimum duration, etc.), for the exception of an adjusted p-value threshold (p<0.025).

Replay events were classified as rest or awake local replay. Awake replay was defined as replay events occurring while the animals were running on either of the tracks and classified as local if the content of the replay event reflected the current track on which the animal was running. Replay events detected during sleep or rest periods within the rest pot were classified as rest replay events.

## Replay events scoring and significance

Replay events were scored using weighted correlation between decoded posterior probabilities across position and time.

Weighted mean:

$$m\left(x;prob\right) = \frac{\sum\limits_{i=1}^{M}\sum\limits_{j=1}^{N}prob_{ij}x_i}{\sum\limits_{i=1}^{M}\sum\limits_{j=1}^{N}prob_{ij}}$$

Weighted covariance:

$$cov\left(x,t;prob\right) = \frac{\sum\limits_{i=1}^{M}\sum\limits_{j=1}^{N}prob_{ij}\left(x_i-m\left(x;prob\right)\right)\left(t_j-m\left(t;prob\right)\right)}{\sum\limits_{i=1}^{M}\sum\limits_{j=1}^{N}prob_{ij}}$$

Weighted correlation:

$$corr\left(x,t;prob\right) = \frac{cov\left(x,t;prob\right)}{\sqrt{cov\left(x,x;prob\right)cov\left(t,t;prob\right)}}$$

where $x_i$ is the ith position bin, $t_j$ is the jth time bin, and $prob_{ij}$ is the probability at the position bin $i$ and time bin $j$.

The statistical significance of the weighted correlation for each candidate replay event was assessed by comparison with three different 1000 shuffle distributions:

1. Spike train circular shift, in which the spike count vectors for each cell were independently circularly shifted in time within each replay event, prior to decoding.
2. Place field shift, in which each ratemap was circularly shifted in space by a random amount of position bins prior to decoding.
3. Circular shift of position, in which posterior probability vectors for each time bin were independently circularly shifted by a random amount.

If the score of the candidate event was greater than the 95th percentile of the distribution for all three shuffles, then the event was considered to be significant. In a few occasions, replay events were found to be significant for both tracks. For our regression analysis (*Figures 1–3*), those events were assigned to one of the tracks by computing the 'Bayesian bias' score for each track. Each score was calculated as the sum of the posterior probability matrix for one track, normalized by the total sum across both tracks. To assign the replay event to a specific track, the Bayesian bias score had to be greater than 60%, otherwise the event was discarded. For our log odds analysis (*Figures 4 and 5*), replay events detected as significant for both tracks were not used in the analysis to avoid our log odds analysis being confounded by our replay event selection method. Out of a total of 5396 significant replay events (across PRE, RUN, and POST), we found a total of 356 replay events detected as significant on both tracks simultaneously, a ratio of 0.066.

## Reinstatement of rate modulation analysis

Linear regression was used to measure the reinstatement of rate modulation between place fields on the track and replay events (MATLAB function *fitlm*). For each place cell active on both tracks, rate modulation on the track was measured by calculating (1) the difference in peak in-field firing rate across tracks and (2) the difference in the area under the curve (AUC; MATLAB function *trapz*). Place fields with a peak in-field firing rate <1 Hz on either track were excluded. As a control, a variation of the analysis was done by excluding all cells with multiple place fields on the same track. Multiple peaks for a single cell were considered to be different place fields when their peak firing rate was >1 Hz, each place field width was over two spatial bins (20 cm), and the distance between the peaks was of four spatial bins (40 cm).

Rate modulation during replay events was calculated as the change in each cell's firing activity between the replay events encoding for each track. Included cells had to participate in at least one replay event for each track. The firing rate change for a given neuron was calculated only using replay events where that neuron fired one or more spikes, and was measured as (1) the difference in the average peak instantaneous rate across tracks, (2) the difference in the median rate across replay events, and (3) the difference in the mean number of spikes across replay events.

To obtain the peak instantaneous rate of each event, each cell's spike train was binned into 1 ms steps, filtered with a 100-ms-long Gaussian window with σ = 20 ms (MATLAB command *filter*), and the peak amplitude of the resulting signal measured.

## Controls for replay detection analysis

Two types of shuffles were applied to the data before repeating the detection of replay events using Bayesian decoding: a track rate shuffle and replay rate shuffle. The track rate shuffle consisted in the randomization of the firing rate of the place fields within a track, while the replay rate shuffle randomized the cells' firing rate during the replay events. For both shuffles, each place cell's firing rate was scaled based on a value randomly drawn from a distribution of firing rates obtained from the analyzed place cells on the track (for the track rate shuffle) or during the replay events (for the replay rate shuffle). The shuffled data was then used to repeat the detection of replay events, and the newly identified significant replay events were used in a linear regression comparing their original spike content (instead of the shuffled one) against the original differences in the peak in-field rate.

## Remapping classification

A bootstrapping procedure was used to identify the rate and place modulation properties of each cell. Modulation was defined when the track difference in a place field's specific parameter (e.g., peak in-field firing rate or peak location) exceeded the intrinsic variability of such parameter on a single track. The distribution of each parameter was computed for both tracks by calculating firing ratemaps from a randomly sampled N out of N laps (1000 iterations, with replacement). The median value of a track distribution was then compared to the 5th/95th percentiles of the other track, and vice versa. If either median exceeded either percentile, the cell was classified as being modulated between tracks for that parameter.

## Sequenceless Bayesian decoding

The sequenceless Bayesian decoding approach used here was modified from a previous method described by *Carey et al., 2019*. Only cells with stable place fields on the track in the first and second half of the behavioral episode (peak in-field firing rate >1 Hz) and replay events significant for a single track were included in our sequenceless decoding analysis.

Before Bayesian decoding, ratemaps for Track 1 and Track 2 were concatenated as a single matrix [nCell × (nPosition(T1) + nPositions(T2))]. Unless otherwise mentioned, 20 ms time bins and 10 cm position bins were used for decoding. After decoding, the posterior probabilities for each time bin were normalized across all position bins from both tracks to sum to 1. Then, the summed posterior probability across all time bins within each replay event was computed, and the probabilistic bias towards Track 1 or Track 2 for each replay event was quantified as $\log\left(\frac{\sum Prob\left(Track_1\right)}{\sum Prob\left(Track_2\right)}\right)$ (log odds), where $\sum Prob\left(Track_1\right)$ and $\sum Prob\left(Track_2\right)$ referred to the summed posterior probabilities for Track 1 and Track 2, respectively. To remove any intrinsic probabilistic bias, each raw log odds calculation was then z-scored relative to a shuffled log odds distribution (ratemap track label shuffle), obtained by randomly permuting each cell's Track 1 and Track 2 ratemaps 1000 times for each replay event. The permutation was done by either keeping a cell's ratemap unaltered or performing a between track swap.

To test the decoding when only using place or rate information, the original data were manipulated as the following:

- Rate fixed manipulation: Rate information was fixed both by rescaling each cell's peak in-field firing rate to the mean value across both tracks and by rescaling its spike count in each replay event to the average firing rate across all replay events.
- Place removed manipulation: The place (and sequence) information was removed by decoding with only a single position bin, which encompassed the entire track (mean firing rate on each track when speed is above 5 cm/s) and a single time bin, which spanned the entire replay event.

Two negative controls were used to disrupt both rate and place information:

- Rate fixed place randomized manipulation: After the rate fixed manipulation, place information was randomized using a cell ID shuffle in which all cells were randomly reassigned to a different cell's ratemap for each replay event.
- Place removed rate randomized manipulation: After place removed manipulation, rate information was randomized by scaling the firing rate on each track based on a value randomly drawn from a distribution of firing rates obtained from the analyzed place cells on the track.

**Table 4.** Summary statistics for the z-scored log odds difference from 10 negative control simulations for PRE.

The negative control manipulation used for main result was selected (indicated in bold) based on the median difference value out of 10 simulations.

|  | Rate fixed place randomized | | | Place removed rate randomized | | |
|---|---|---|---|---|---|---|
|  | Mean difference | SE | p-Value | Mean difference | SE | p-Value |
| Sim 1 | 0.18 | 0.06 | 0.002 | 0.18 | 0.06 | 0.002 |
| Sim 2 | –0.08 | 0.12 | 0.79 | 0.01 | 0.05 | 0.79 |
| Sim 3 | –0.01 | 0.12 | 0.52 | –0.03 | 0.06 | 0.79 |
| Sim 4 | 0.05 | 0.06 | 0.79 | –0.14 | 0.07 | 0.99 |
| Sim 5 | –0.03 | 0.04 | 0.52 | –0.02 | 0.05 | 0.57 |
| Sim 6 | 0.03 | 0.09 | 0.52 | –0.08 | 0.09 | 0.76 |
| Sim 7 | –0.02 | 0.13 | 0.52 | –0.03 | 0.05 | 0.48 |
| **Sim 8** | **0.03** | **0.06** | **0.24** | **–0.02** | **0.05** | **0.63** |
| Sim 9 | 0.08 | 0.11 | 0.06 | 0.01 | 0.05 | 0.052 |
| Sim 10 | –0.03 | 0.09 | 0.52 | –0.03 | 0.05 | 0.26 |

Given the inherent random nature of our shuffling procedure, it was important to ensure that the result of a single simulation was reproducible. Thus, for both negative control manipulations, we ran our simulations 10 times. We observed that 9/10 times the results were not significant (rank-sum test, p>0.05, uncorrected). The data presented in *Figure 5* was selected from the negative control simulation nearest the median log odd difference (out of 10 simulations) (*Table 4*).

## ROC analysis

The 'ground truth' track identity for each replay event was assigned based on the sequenceness of the replay, which was determined by comparing the event's weighted correlation score to three shuffled distributions. The binary discriminability of a replay event's identity (Track 1 or Track 2) during PRE, POST, or RUN was quantified using the ROC curve. The ROC curve was constructed based on a range of true- and false-positive rates obtained by systematically shifting the discrimination threshold along the z-scored log odds distributions for both Track 1 and Track 2 replay events. A trapezoidal approximation was used to estimate the area under each ROC curve (MATLAB function *perfcurve* for both constructing ROC curve and estimating AUC). The bootstrapped distributions of ROC curves and the area under the corresponding ROC curves were produced by resampling with replacement 1000 times.

## Statistics

### Z-scored log odds significance

The statistical significance of the mean difference between Track 1 and Track 2 z-scored log odds is quantified using a one-tailed rank-sum test in which mean z-scored log odds track differences across sessions was compared to the mean shuffled differences across sessions. The shuffled difference distribution for each session was obtained by resampling replay events 1000 times with replacement and assigning their track identities randomly (referred to as replay track identity shuffle). The mean z-scored log odds difference was considered significantly different from the shuffled distribution at the significance level of 0.05.

### Bootstrapping for ROC significance

To determine whether rate and/or place representation manipulations significantly changed the AUC of the ROC curves, we calculated the confidence interval for the difference between each condition's bootstrapped AUC distribution and that of the original data. This was repeated in an additional analysis by replacing the original data by one of the negative controls (rate fixed place randomized

manipulation and place removed rate randomized manipulation). The mean difference between two bootstrapped AUC distributions were only considered statistically significant when the 95% confidence interval did not overlap with 0.

### Delong test for ROC

To determine whether rate and/or place manipulations statistically significantly changed the AUC of ROC curves within the same behavioral epochs, the AUC of both manipulations (i.e., rate fixed or place removed) were compared with the AUC of the original data or their corresponding negative controls (i.e., rate fixed vs. rate fixed place randomized and place removed vs. place removed rate randomized). The DeLong test (*Sun and Xu, 2014*) was employed to perform pairwise comparisons between two ROC curves. The DeLong variance–covariance matrix for two ROC curves was computed using MATLAB-based algorithm (https://github.com/PamixSun/DeLongUI; *Sun, 2015*) developed by *Sun and Xu, 2014*.

A z-score was calculated based on the following equation:

$$z = \frac{|AUC_A - AUC_B|}{\sqrt{var(A) + var(B) - 2cov(A,B)}}$$

where AUC is the area under the ROC curve, var(X) is the variance of ROC curves from a 1000 bootstrapped distribution, and cov(X) is the covariance of ROC curves from both 1000 bootstrapped distributions.

The z-score for each pair of comparisons was subsequently used to calculate the two-tailed p-value. Two AUC distributions were considered significantly different when $p \leq 0.05$.

All analyses were carried out using MATLAB (MathWorks, Natick, MA).

# Acknowledgements

We thank Yu Qian, Sophie Renaudineau, and Julieta Campi for technical assistance; members of the Bendor Lab for valuable discussion; and Aman Saleem and Caswell Barry for their comments on the manuscript. This work was supported by the Biotechnology and Biological Sciences Research Council scholarship (BB/M009513/1) (MTir), the European Research Council starter grant (CHIME) (DB), the Human Frontiers Science Program Young Investigator Award (RGY0067/2016) (DB), and the Biotechnology and Biological Sciences Research Council Research grant (BB/T005475/1) (DB). The Titan Xp used for this research was donated by the NVIDIA Corporation

# Additional information

### Funding

| Funder | Grant reference number | Author |
| --- | --- | --- |
| Biotechnology and Biological Sciences Research Council | BB/M009513/1 | Margot Tirole |
| Biotechnology and Biological Sciences Research Council | BB/T005475/1 | Daniel Bendor |
| European Research Council | CHIME | Daniel Bendor |
| Human Frontier Science Program | RGY0067/2016 | Daniel Bendor |

The funders had no role in study design, data collection and interpretation, or the decision to submit the work for publication.

### Author contributions

Margot Tirole, Conceptualization, Software, Formal analysis, Funding acquisition, Investigation, Visualization, Methodology, Writing – original draft, Writing – review and editing, The order of shared first author position does not signify and should not be interpreted to suggest any difference in the

work and commitment to the project by either of the individual first authors, and was determined by a coin-flip. Both MTir and MHG contributed equally and agree to reserve the right to list their name first in their CV; Marta Huelin Gorriz, Conceptualization, Software, Formal analysis, Investigation, Visualization, Methodology, Writing – original draft, Writing – review and editing; Masahiro Takigawa, Conceptualization, Software, Formal analysis, Visualization, Methodology, Writing – original draft, Writing – review and editing; Lilia Kukovska, Visualization, Methodology, Writing – review and editing; Daniel Bendor, Conceptualization, Software, Supervision, Funding acquisition, Methodology, Writing – review and editing

### Author ORCIDs
Margot Tirole http://orcid.org/0000-0003-0674-6690
Marta Huelin Gorriz http://orcid.org/0000-0002-0281-0627
Masahiro Takigawa http://orcid.org/0000-0002-0162-9017
Daniel Bendor http://orcid.org/0000-0001-6621-793X

### Ethics
All experimental procedures and post operative care were approved and carried out in accordance with the UK Home Office, subject to the restrictions and provisions contained within the Animal Scientific Procedures Act of 1986. Experiments were conducted under PPL P61EA6A72 (Bendor).

### Decision letter and Author response
Decision letter https://doi.org/10.7554/eLife.79031.sa1
Author response https://doi.org/10.7554/eLife.79031.sa2

---

## Additional files

### Supplementary files
• MDAR checklist

### Data availability
The data used in this manuscript are available on DRYAD [https://doi.org/10.5061/dryad.ksn02v76h]. All custom-written code is available on GitHub (https://github.com/bendor-lab/Elife_Tirole_Huelin_Gorriz_2022 copy archived at swh:1:rev:44ecf4275c2a7ca33dda6b67f11b4e854c3123e9).

The following dataset was generated:

| Author(s) | Year | Dataset title | Dataset URL | Database and Identifier |
| --- | --- | --- | --- | --- |
| Tirole M, Huelin Gorriz M, Bendor D | 2022 | data from: Experience-driven rate modulation is reinstated during hippocampal replay | https://doi.org/10.5061/dryad.ksn02v76h | Dryad Digital Repository, 10.5061/dryad.ksn02v76h |

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
