## [Editor Report]

The hippocampal cells that comprise the place cell map for the most part ‘remap’ between different environments – they change their preferred firing locations and rates. This article poses an important question about offline reactivation that has not been explicitly tested: are differences in firing rate preserved during sequential, temporally compressed offline replay events? They find that yes, individual neurons show context-specific firing rates during replay, an important finding and a confirmation of critical theoretical foundations in the field of learning and memory. The evidence is convincing, with good support for the claims. At the same time, this demonstration hinges on some relatively subtle methodological points specific to replay detection, and thus serves as an invitation to the field to further explore the precise structure of context-specific offline activity.

---

## [Decision Letter]

**Decision letter after peer review:**

Thank you for submitting your article "Experience-driven rate modulation is reinstated during hippocampal replay" for consideration by *eLife*. Your article has been reviewed by 3 peer reviewers, including Caleb Kemere as the Reviewing Editor and Reviewer #1, and the evaluation has been overseen by Laura Colgin as the Senior Editor.

Essential revisions:

1) In their discussion, the reviewers agreed that there is prior work that demonstrates the concept that contextual information present as a rate-modulation in the CA1 place-code is also present in replay. It is important that your work cite these prior efforts. In particular, the work of Takahashi, 2015, Farooq et al. 2019, and Gupta et al. 2010 merit further consideration in terms of rate modulation, and Karlsson et al. 2009 in terms of replaying multiple environments. The reviewers were convinced that your work is sufficiently different from these in that it presents activity recorded in different mazes in subsequent sleep, but it is critical to place it among related works.

2) The reviewers agreed that a strength of the work was its focus on a particular question. In discussion, we concluded it was unfair and unnecessary to ask for substantial new analysis. That said, if the primary question the paper is answering is "Are the firing rate differences that mark maps of distinct experiences maintained during replay?" the paper would be stronger if it could report the extent to which this effect is general and not merely a characteristic of a small fraction of events that are already extremely distinct (i.e. "replays" vs "ripples"). To that end, at a minimum please report what fraction of ripples (i.e., putative replay events) your subsequent analyses capture. In addition, if it's relevant, it might also be worth reporting the sensitivity of the analyses to the significance threshold used for replay detection (e.g., what would happen if it dropped from 95% to 90%).

*Reviewer #1 (Recommendations for the authors):*

My two requests are that they repeat their analyses (a) for all ripples or population burst events and (b) repeat the analyses using all neurons.

Figure 1B. It is unclear what the decoded result looks like when only a few decoded positions appeared in the figures in the last row of the first column and the second row of the second column.

Page 10: "We found that place information alone was sufficient…". I think that this is not really an accurate statement. I think what the authors mean, and what would be more clear is "We found that the ensemble co-activity – i.e., which cells were active in a decoding bin – was sufficient…"

Similarly, "Next, we selectively removed place information without altering rate information…". I also don't love this formulation, though it is more true. I would simply say, "we removed the sequential order of firing".

Page 12. Discussion, second paragraph, the statement of the first key advance is confusing. Does it mean sequential information is better for replay detection and rate modulation is more suitable for context determination?

Page 19. The first equation at the top, how is P(x) defined?

Page 19. Replay events scoring and significance. In the weighted covariance equation, what is y?

Why is "Bayesian bias" score computed as the sum of the posterior probability matrix rather than decoded position likelihood?

Page 20. Reinstatement of rate modulation analysis

Paragraph 2, last two lines, do "replay events" in (2) and (3) include all replay events or only events when the cell fired?

*Reviewer #3 (Recommendations for the authors):*

The following methodological details need to be elaborated/clarified for replication purposes:

– It is unclear how much prior training the rats had. The first sentence of Results says they were trained to run, but this training is not described. While the tracks are novel, for replication purposes, it should be clear if they have had prior experience on any linear tracks prior to recording/implantation and how similar/distinct those training tracks were to the novel tracks used during the recordings.

– In the criteria for defining a place field, it is unclear how "stable spiking activity across the first half and second half" is quantified. The way in which the PPV is calculated is clear, but it is not clear what criteria defined "stable" vs. "unstable."

– When candidate replay events are speed-filtered, I'm guessing the authors mean that they kept only events when the rat's velocity was less than 5 cm/s. The text implies that they kept events with >5 cm/s.

– "In a few occasions, replay events were found to be significant for both tracks." For interpretation, please list the number (and percent) of such events.

In addition, while not necessary in my opinion for publication, I think analyzing direction-specific place fields may provide an even stronger argument (as you would effectively have 4 tracks for comparison rather than just 2).

---

## [Author Response]

Essential revisions:1) In their discussion, the reviewers agreed that there is prior work that demonstrates the concept that contextual information present as a rate-modulation in the CA1 place-code is also present in replay. It is important that your work cite these prior efforts. In particular, the work of Takahashi, 2015, Farooq et al. 2019, and Gupta et al. 2010 merit further consideration in terms of rate modulation, and Karlsson et al. 2009 in terms of replaying multiple environments. The reviewers were convinced that your work is sufficiently different from these in that it presents activity recorded in different mazes in subsequent sleep, but it is critical to place it among related works.

Thank you for these additional citations, which have now been integrated into an expanded intro (page 2) and discussion (page 14-15) of our revised manuscript.

2) The reviewers agreed that a strength of the work was its focus on a particular question. In discussion, we concluded it was unfair and unnecessary to ask for substantial new analysis. That said, if the primary question the paper is answering is "Are the firing rate differences that mark maps of distinct experiences maintained during replay?" the paper would be stronger if it could report the extent to which this effect is general and not merely a characteristic of a small fraction of events that are already extremely distinct (i.e. "replays" vs "ripples"). To that end, at a minimum please report what fraction of ripples (i.e., putative replay events) your subsequent analyses capture. In addition, if it's relevant, it might also be worth reporting the sensitivity of the analyses to the significance threshold used for replay detection (e.g., what would happen if it dropped from 95% to 90%).

We agree with the reviewers that it is important to demonstrate that the main observations are not due to a small subset of replay events. In the revised manuscript we provide a new supplementary figure (Figure 1—figure supplement 6), that uses several less strict criteria for comparison– (1) a p value threshold for replay detection of 0.1 (instead of 0.05) [as requested], (2) two shuffles for replay detection rather than three, and (3) no minimum ripple power criteria. The fraction of significant replay events out of the total number of candidate events is also now provided for the original dataset (page 2 of revised manuscript) along with these three less stringent criteria for replay detection (Figure 1—figure supplement 6 legend). In addition, we felt it would be useful for comparison to also examine altering the criteria for replay detection in the opposite direction- does using an even stricter criteria for replay detection, where we now have presumably “better quality” replay events, show a more pronounced effect. Thus, we restricted the replay detection criteria to P<0.01 as an additional subpanel of the supplementary figure (Figure 1—figure supplement 6) for comparison. Our observation is that across all criteria, there is a significant regression between place field rate difference and replay rate difference (between tracks), however it becomes more pronounced as the criteria for replay becomes stricter.

We also now provide the replay detection ratio (significant replay events / total number of candidate replay events) for the different detection criteria (original, P<0.1, P<0.01, 2 shuffle, no ripple threshold) in the figure legend of Figure 1—figure supplement 6 and the main text.

Reviewer #1 (Recommendations for the authors):My two requests are that they repeat their analyses (a) for all ripples or population burst events and (b) repeat the analyses using all neurons.

We have repeated our analysis for all neurons (as discussed above) in (Figure 1—figure supplement 6). We have made our replay detection criteria less stringent in three ways (discussed above), which has now been integrated into Figure 1—figure supplement 6.

Figure 1B. It is unclear what the decoded result looks like when only a few decoded positions appeared in the figures in the last row of the first column and the second row of the second column.

The reason that only a few decoded positions appear in this plot, is that our decoding method normalizes the posterior probability over both tracks, so if the replay is only of track 1, we do not observe high values for the posterior probability in the majority for the decoded replay of track 2. This approach of normalized posterior probability across both tracks is key to using the log odds method in Figure 4 and 5. We have clarified this point in the Figure 1 legend of the revised manuscript-

“Because posterior probabilities are normalized across both tracks, decoded positions during behavior and replay can have substantially higher likelihood values on one track.”

Page 10: "We found that place information alone was sufficient…". I think that this is not really an accurate statement. I think what the authors mean, and what would be more clear is "We found that the ensemble co-activity – i.e., which cells were active in a decoding bin – was sufficient…".

We agree that co-activity makes more sense, however because we did not specifically test co-activity (we removed only removed rate information), we have modified the sentence as follows:

“We found that the remaining information, arising most likely from ensemble co-activity – i.e., which cells were active in a decoding bin – was sufficient…”.

Similarly, "Next, we selectively removed place information without altering rate information…". I also don't love this formulation, though it is more true. I would simply say, "we removed the sequential order of firing".

We have modified this sentence to reflect this comment as follows:

“we selectively removed the sequential spatial and temporal relationship between neighboring place fields without altering rate information…”.

Page 12. Discussion, second paragraph, the statement of the first key advance is confusing. Does it mean sequential information is better for replay detection and rate modulation is more suitable for context determination?

To help clarify this statement, we have rephrased it as follows- “First is our analysis of replay rather than reactivation, which uses the sequential pattern of firing across place cells for detection, in turn providing significantly more statistical power than using a sequenceless based reactivation approach”.

Sequential information is better for replay detection simply because replay is defined as a sequence, in contrast to reactivation. We don’t know if sequence detection or sequenceless decoding is better for context determination, but they both can be used, which is the basis of our cross-validation approach in figures 4 and 5.

Page 19. The first equation at the top, how is P(x) defined?

Thank you for catching this. We applied a naïve Bayes decoder so the prior P(x) is set to 1. We have removed the P(x) term from the equation in the revised manuscript as it is unnecessary. The corrected equation is belowP(x|n)=C(∏i=1Nfi(x)ni)exp⁡(−τ∑i=1Nfi(x))

Page 19. Replay events scoring and significance. In the weighted covariance equation, what is y?

Thanks for catching this error. Y should be T, and has now been corrected. The covariance is calculated over time and position bins.

Why is "Bayesian bias" score computed as the sum of the posterior probability matrix rather than decoded position likelihood?

We have further clarified this section of the methods section in the revised manuscript. If a replay event is detected as significant on both tracks, we examine the matrix of decoded position-time bins, and sum across time and position for each track. This is essentially the approach used in the log-odds analysis without the track id shuffling procedure. Because the posterior probabilities across all position bins (on both tracks) within a single time bin must sum to 1, this approach measures how much of the total probability is shifted towards one track. Comparing the maximum likelihood between each track may ignore a track that has a larger sum of posterior probabilities, just distributed over more position bins.

Page 20. Reinstatement of rate modulation analysisParagraph 2, last two lines, do "replay events" in (2) and (3) include all replay events or only events when the cell fired?

We have clarified this now in the text as follows- “The firing rate change for a given neuron was calculated only using replay events where that neuron fired one or more spikes, and was measured as (1) …”.

Reviewer #3 (Recommendations for the authors):The following methodological details need to be elaborated/clarified for replication purposes:– It is unclear how much prior training the rats had. The first sentence of Results says they were trained to run, but this training is not described. While the tracks are novel, for replication purposes, it should be clear if they have had prior experience on any linear tracks prior to recording/implantation and how similar/distinct those training tracks were to the novel tracks used during the recordings.

We have added this statement in our methods section on page 18 to clarify this point

“…Prior to the start of recordings, rats were trained for approximately two days, 30 min each, to run back and forth on a linear track with reward delivered at each end. Training occurred in a different room and track from the one used during the recordings.”

“…The linear tracks used in every experiment were designed to be novel to the animal. To accomplish this, the shape of the tracks was changed between recording sessions and their surfaces covered with different textured fabrics. In each session, the room was surrounded by black curtains with different high contrast visual cues. The tracks were separated using view-obstructing dividers.“

– In the criteria for defining a place field, it is unclear how "stable spiking activity across the first half and second half" is quantified. The way in which the PPV is calculated is clear, but it is not clear what criteria defined "stable" vs. "unstable."

We required that a place field had a peak firing rate>1spk/s for both the first and second half of the exploration of the track to be considered stable.

We have modified this description in the methods section on page as follows: “A principal cell was classified as a place cell if it had a minimum peak firing rate that was >1Hz in its unsmoothed ratemap. Furthermore, place cells were required to have stable place fields, defined as a minimum peak firing rate > 1 Hz for both the first half and second half of the RUN session“.

– When candidate replay events are speed-filtered, I'm guessing the authors mean that they kept only events when the rat's velocity was less than 5 cm/s. The text implies that they kept events with >5 cm/s.

thank you for catching this. We have changed the text now to “the animal’s speed was required to be less than 5 cm/s”.

– "In a few occasions, replay events were found to be significant for both tracks." For interpretation, please list the number (and percent) of such events.

We found that approximately 6.6% of significant replay events were detected as significant on both tracks. We have indicated this in the Methods section on page 22 as follows:

“Out of a total of 5396 significant replay events (across PRE, RUN, and POST), we found a total of 356 replay events detected as significant on both tracks simultaneously, a ratio of 0.066.”

In addition, while not necessary in my opinion for publication, I think analyzing direction-specific place fields may provide an even stronger argument (as you would effectively have 4 tracks for comparison rather than just 2).

We agree that this is an interesting analysis to do, albeit slightly beyond the scope of our main goals and a similar analysis (direction selective place fields) was reported by Schwindel, Navratilova, Ali, Tatsuno, and McNaughton, 2016. One potential issue with direction-specific place fields is that they either fire only in one direction (similar to neurons in our experiment that only fire on one track) or fire at the same location for both directions (only changing their firing rate). If all cells were to purely rate remap, we would lose the ability to use the sequence of place cell activation (rate independent) as a “ground truth” to cross-validate replay content. Conversely, cells that fire only in one direction are prone to confounds discussed above (see response to reviewer #1).